# Response of Siberian Cranes (*Grus leucogeranus*) to Hydrological Changes and the Availability of Foraging Habitat at Various Water Levels in Poyang Lake

**DOI:** 10.3390/ani14020234

**Published:** 2024-01-11

**Authors:** Mingqin Shao, Jianying Wang, Hongxiu Ding, Fucheng Yang

**Affiliations:** School of Life Sciences, Jiangxi Normal University, Nanchang 330022, China; jianying@jxnu.edu.cn (J.W.); yang-fc@jxnu.edu.cn (F.Y.)

**Keywords:** Siberian crane, habitat suitability index, MaxEnt, Poyang Lake, inundated area, hydrological dynamics

## Abstract

**Simple Summary:**

The critically endangered Siberian crane (*Grus leucogeranus*) relies on Poyang Lake’s wetlands in China for wintering. Rapid changes in these wetlands, driven by climate shifts and human activities, affect habitat and food availability for these birds. Our study investigated Poyang Lake’s hydrological dynamics over the past two decades and assessed the impact of related variables on crane populations. Employing advanced modeling, we predicted how varying water levels influence suitable habitats for Siberian cranes. Our findings revealed that a summer-inundated area exerted a notable influence on population fluctuations in crane species in the natural habitats of Poyang Lake. The scarcity of food, resulting from summer floods, compels cranes to adapt by exploring alternative food sources and new foraging grounds, including artificial habitats. During the dry season, the size of the inundated area primarily impacts the food availability for Siberian cranes, thereby influencing both their population size and distribution pattern, and a moderate water level of 8–10 m provides the highest amount of good habitat. Persistent winter droughts exacerbate the unsuitability of habitats, and the shortage of food in natural habitats may become a recurring occurrence in the future. This study emphasizes the importance of managing both natural and artificial habitats to support Siberian crane conservation amid environmental changes.

**Abstract:**

To assess the Siberian crane (*Grus leucogeranus*)’s response to changing water levels and habitat quality at Poyang Lake, we analyzed the lake’s hydrological trends over the past two decades with the Mann-Kendall and Sen slope methods. Additionally, we explored the link between the crane population size and hydrological conditions at the lake from 2011 to 2019. Meanwhile, five environmental factors, including habitat type, distance from shallow lakes, human footprint index, elevation and normalized vegetation index were selected, and the distribution patterns of suitable habitats for the Siberian crane under 10 water level gradients with intervals of about 1 m (5.3–14.2 m) were simulated by using an improved habitat suitability index model that determines the weights of evaluating factors based on the MaxEnt model. The results showed that the overall trend of the inundated area in Poyang Lake was shrinking in the last 20 years, with a significant increase in the area of exposed floodland during the early wintering period (Z = −2.26). The prolonged drought resulting from this will force vegetation succession, thereby diminishing the food resources for cranes in their natural habitat. The mean inundated area in June demonstrated a significant negative correlation with the population of Siberian cranes in natural habitats (*r* = −0.75, *p* = 0.02). Shortage of the Siberian crane-preferred *Vallisneria* tuber due to June flooding was the primary driver of the crane’s altered foraging strategy and habitat shift. In years with relatively normal June inundation, indicating abundant *Vallisneria* resources, the relationship between the inundated area during the dry season and the crane population fit well, with a quadratic curve (*R*^2^ = 0.92, *p* = 0.02). The dry season’s inundated area primarily affected the crane population and distribution pattern by influencing the availability of food resources, and both excessive and insufficient inundation areas were unfavorable for crane survival. The modeling results for habitat suitability indicated that as the water level decreased, the trend of the area of good habitat for the Siberian crane showed an inverted bell shape, peaking at a water level of 8.8 m, with optimal conditions occurring between 8 and 10 m. The combined effects of climate and human activities have made the shortage of food resources in Poyang Lake the new normal. The degradation of natural habitats has led to a decline in the quality of Siberian crane habitats, and artificial habitats can only be used as refuges to a certain extent. Thus, formulating strategies to restore natural habitats and enhance the management of artificial habitats is crucial for the conservation efforts of Siberian cranes.

## 1. Introduction

Wetlands, serving as an intermediate ecological zone between terrestrial and aquatic environments, represent one of the three major ecosystems globally. They play crucial ecological roles, including in water conservation, climate regulation, flood and drought control, and biodiversity protection [1,2]. Waterbirds, constituting a vital component of wetland ecosystems, serve as key indicators of wetland health. The size and distribution of waterbird populations are directly influenced by the ecological processes within wetlands [3,4]. Unfortunately, wetlands are facing a rapid global decline, leading to a reduction in area and function, posing a severe threat to the survival of numerous endangered waterbird species [1,5].

Poyang Lake, the largest connected lake of the Yangtze River in China, holds international significance as a crucial wintering site for migratory birds, hosting over 700,000 wintering waterbirds, including the critically endangered Siberian crane (*Grus leucogeranus*) (International Union for Conservation of Nature, IUCN, 2018) [6,7]. The Siberian crane, a large wading bird highly dependent on wetland habitats, comprises approximately 5600 individuals globally, with the largest population, the eastern population, predominantly wintering at Poyang Lake [6,8,9]. In many years, more than 95% of the global Siberian crane population has been found wintering at Poyang Lake [10]. Poyang Lake experiences significant annual cyclical fluctuations in water levels, creating a diverse landscape of wetland vegetation [3,11]. During the dry season, the depths of dish-shaped lakes reach levels suitable for Siberian crane foraging (less than 45 cm). Shallow lakes with abundant underground tubers, particularly those of submerged vegetation like *Vallisneria*, serve as the primary food source for Siberian cranes. These areas also provide an ideal foraging environment due to their distance from human disturbances [12,13].

However, under the influence of climate change and human activities such as sand mining in the basin, the hydrological characteristics of Poyang Lake have changed dramatically. These alterations have involved the advancement and prolongation of the dry season, coupled with an increased frequency of floods and droughts [14,15]. Consequently, the type of wetland vegetation and the pattern of succession have experienced drastic alterations [3,7]. These changes pose a serious threat to the survival of Siberian cranes and other phytophagous waterfowl. It was first observed that the Siberian crane abandoned its preferred shallow water habitats and chose to forage the roots of *Potentilla limprichtii* and the bulbs of *Tulipa edulis* in grassland at the end of 2010 [12]. During the wintering period of 2015–2016, Siberian cranes left their natural wetlands and migrated at a large scale to artificial habitats such as farmlands and lotus root ponds [13]. Since then, some Siberian cranes have been continuously counted in the Yellow River Delta 1000 km away from Poyang Lake, and new records of Siberian cranes overwintering have even appeared in areas where Siberian cranes have never been before, such as Fujian and Guangdong [8]. Food abundance is a key factor in habitat selection by the Siberian crane [9]. Floods in the spring and summer of 2010 and 2015–2016 collapsed the density of *Vallisneria* tubers in Poyang Lake, and the degradation of habitats and shortage of staple foods are considered to have been the key drivers of the Siberian crane’s food composition change, habitat shift and population dispersal [13]. Therefore, identifying how to provide high-quality habitats for the Siberian crane through effective wetland management is a key issue for maintaining the Siberian crane population.

Habitat suitability evaluation is a crucial tool for species conservation and habitat management [16]. The Habitat Suitability Index (*HSI*) model, a widely used method, relies on a Geographic Information Systems (GIS) to analyze species-environment interactions, assigning a suitability index ranging from 0 to 1 to quantify habitat quality in the study area [9,17]. However, this model’s construction depends on expert experience and lacks support from ecological niche theory, leading to potential deviations between output results and the actual suitable habitats of the species [18,19]. An alternative model for habitat evaluation is MaxEnt, a machine learning model that estimates species’ ecological needs based on distribution points and environmental variables, predicting potentially suitable habitats [20,21,22]. Nevertheless, obtaining accurate species distribution data poses challenges, making it less suitable for simulating suitable habitats across multiple time points. For the Siberian crane, a species inhabiting highly variable environments, habitat evaluation under a single temporal or environmental condition may not suffice for habitat management needs [23]. Therefore, a coupled modeling approach is essential to constructing a set of habitat suitability evaluation methods applicable to various environmental conditions.

In this study, the primary objective is to comprehend the changing characteristics of Siberian crane habitat quality over the last 20 years and to investigate the hydrological mechanisms driving habitat shifts. This analysis is based on the long time series of Poyang Lake’s inundated area data from 2000 to 2020. This study aims to explore the correlation between the population size of Siberian cranes and hydrological characteristics. To determine the spatial distribution patterns of suitable habitats for Siberian cranes during the wintering period under varying water levels, a MaxEnt-HSI coupled model was constructed. This model analyzed the quantitative relationship between the area of suitable habitats for Siberian cranes and water levels, revealing suitable habitats at different water level intervals for the species. The overarching goal is to conduct a comprehensive evaluation of the habitat quality of Siberian cranes through a long-term study of hydrological dynamics and the simulation of suitable habitats in response to water level changes. This research seeks to elucidate the response mechanism of Siberian cranes to hydrological environmental changes, providing a basis for the management of wetland ecosystems and the conservation of Siberian cranes in Poyang Lake.

## 2. Materials and Methods

### 2.1. Study Region

Poyang Lake (24°22′–29°45′ N, 115°47′–116°45′ E) is positioned on the southern bank of the middle and lower reaches of the Yangtze River in northern Jiangxi Province, China (Figure 1). The lake is characterized by two main parts: northern and southern sections. The southern part, serving as the primary lake body, features a wide and shallow surface and receives water from five tributaries: the Ganjiang River, Fu River, Xiushui River, Xinjiang River, and Raohe River. This water eventually exits through the outlet of the lake after storage and accumulation. In contrast, the northern part is narrower with lower terrain and serves as the waterway into the Yangtze River [9,17,24]. Poyang Lake exhibits characteristics typical of a seasonal lake, influenced by the water levels of the “Five Rivers” and the Yangtze River. During the flood season (April–September), substantial water enters the lake, resulting in an inundation area exceeding 3000 km^2^. In contrast, the dry season (October–March) experiences limited precipitation, causing the water level to recede. Consequently, the lake becomes exposed, the water returns to the trough, and the submerged area decreases to less than 1000 km^2^. This seasonal variation defines the hydrological dynamics of Poyang Lake [25,26]. The receding land and water zone during the dry season support diverse wetland vegetation communities and harbor numerous aquatic biological remnants, offering abundant food resources for wintering migratory birds, including the Siberian crane [27]. Recognizing the ecological significance of Poyang Lake, two national nature reserves, namely Poyang Lake National Nature Reserve and Nanji Wetland National Nature Reserve, have been established, along with three provincial nature reserves. These reserves play a vital role in contributing to the protection of wetland ecosystems and the conservation of migratory birds in the region [17].

### 2.2. Data Sources on Hydrology and Crane Population

The inundated area data for Poyang Lake, spanning from 2000 to 2020, were acquired from the “2000–2020 Global Surface Dynamic Water Body Dataset”. This dataset relies on NDVI data (MODIS) with a temporal resolution of 8 days and a spatial resolution of 250 m, accessible on the Science Data Bank website (https://www.scidb.cn/en, accessed on 8 October 2023). This method of surface water mapping has demonstrated superior results compared to alternative water body extraction methods, achieving an overall accuracy of 96% and a kappa coefficient of 0.9 [28]. Within ArcGIS, areas with an attribute value of “1” (indicating water) were extracted and analyzed, resulting in a total of 955 data points. Water level data for Poyang Lake in the years 2021–2022 were obtained from the Jiangxi Provincial Water Resources Department (http://slt.jiangxi.gov.cn/, accessed on 12 October 2020).

The Siberian crane distribution data were collected from various sources, encompassing field positioning by the research group, the “2021–2022 Poyang Lake 10 Rare Migratory Birds Sentinel Observation Dataset”, accessible on the China Scientific Data Platform (http://csdata.org/, accessed on 8 October 2023) [29] and the China Birdwatching Record Center (http://www.birdreport.cn/, accessed on 8 October 2023). Population data for Siberian cranes originate from the natural resource monitoring reports of the Poyang Lake National Nature Reserve [30,31,32,33,34,35,36,37]. The survey is collaboratively conducted by the Poyang Lake National Nature Reserve (PLNNR) and Jiangxi Provincial Wildlife Management Bureau (JWMB) during December to January each year, during a period when the Siberian crane population remains relatively stable. The surveyed areas include Poyang Lake National Nature Reserve, Nanji Wetlands National Nature Reserve, Duchang Migratory Bird Provincial Nature Preserve, Kangshan Migratory Bird Nature Reserve and other primary natural habitats for Siberian cranes, covering nearly the entire Poyang Lake area.

### 2.3. Data Processing 

#### 2.3.1. Abrupt Change Point Test

The inundated area data of Poyang Lake at 8-day intervals over the past 20 years, the inundated area in June, and the maximum, mean and minimum values of the inundated area in the dry season (October–March), were recorded annually since 2000. The trend significance test for the interannual change in the dry season’s inundated area of Poyang Lake was conducted using the Mann-Kendall (MK) trend and mutation test. The MK test [38,39] is a non-parametric statistical method that makes no assumptions about the distribution of data, rendering it suitable for situations where the normal distribution assumption is not met. Moreover, the MK test demonstrates good efficacy even in cases of small sample sizes, making it widely applicable in the analysis of time series trends. The MK trend test is calculated as the following: (1)$$s=\sum_{i\;=\;1}^{n\;-\;1}\sum_{j\;=\;i\;+\;1}^{n}sgn\;(X_{i}-X_{j})$$

In the context where *X_i_* and *X_j_* represent random variables, the given time series *X* is divided into two sets of variables: *X*_1_, *X*_2_, …, *X_i_*, and *X_i+_*_1_, *X_i+_*_2_, …, *X_j_*.
(2)$$Sig\left(X_{j}-X_{i}\right)=\left\{\begin{array}{c} 1\;\;\;if\;\;\;X_{j}-X_{i}>0\\ 0\;\;\;\;if\;\;\;X_{j}-X_{i}=0\\ -1\;\;\;if\;\;\;X_{j}-X_{i}<0 \end{array}\right.$$

The variance of *S*, *Var*(*S*), is given by
(3)Var(S)=n(n−1)(2n+5)/18

The statistical formula *S* > 0, *S* < 0, *S* = 0 for *Z* is
(4)$$Z_{S}=\left\{\begin{array}{c} \frac{S-1}{\sqrt{Var\;(S)}}\;\mathrm{for}\;S>0\\ 0\;\;\;\;\;\;\;\;\;\;\;\;\;\mathrm{for}\;\;S=0\\ \frac{S+1}{\sqrt{Var\;(S)}}\;\mathrm{for}\;S<0 \end{array}\right.$$

If |*Z_S_*| is greater than *Z*_α/2_, where α represents the chosen significance level (*α* = 5% at a 95% confidence level, with *Z*_0.025_ = 1.96), a positive or negative value of *Z* indicates an increasing or decreasing trend, respectively.

The MK mutation test is calculated as follows and the ordinal column structure is
(5)$$s=\sum_{i\;=\;1}^{k}\sum_{j}^{i\;-\;1}a_{ij}k=2,3,4,\ldots ,n$$
(6)$$a_{ij}=\left\{\begin{array}{c} +1\;\;if\;\;x_{i}>x_{j}\\ \;\;0\;\;else \end{array}\right.\;\;(j=1,2,\ldots ,i)$$

Assuming the randomness of the time series, define the statistic:(7)$$UF_{k}=\frac{\left[s_{k}-E(s_{k})\right]}{\sqrt{Var(s_{k})}}(k=1,2,\ldots ,n)$$
where *UF*_1_ = 0, *E(s_k_*) and *Var*(*s_k_*) represent the mean and variance of the cumulative number *S_k_*:(8)$$E(s_{k})=\frac{n(n-1)}{4}$$
(9)$$Var(s_{k})=\frac{n(n-1)(2n+5)}{72}$$

In the reverse order of the time series *X* (*X_n_*, *X_n_*_−1_, …, *X*_1_), repeat the above process. Let *K* = *n*, *n* − 1, …, 1, where *UB*_1_ = 0 and *UB_k_ =* −*UF_k_*.

Set the significance level α = 0.05, with the critical value *U*_0.05_ = ±1.96. Draw *UF_k_* and *UB_k_* curves, where *UF_K_* values greater than 0 indicate an upward trend, values less than 0 indicate a declining trend, and exceeding the critical value indicates a significant upward or downward trend. When the curves intersect between critical lines, the moment of crossing signifies the onset of mutation.

According to the change trend of the inundated area in the dry season, define the stage between the maximum and minimum inundated areas as the receding stage. The area of water receding every eight days during this phase is termed the receding rate, and the annual decline rate of inundated area is calculated using Sen’s slope trend test. The significance of the interannual change trend is assessed by the *Z* value of the MK test.

#### 2.3.2. Correlation between Siberian Crane Population Size and Hydrological Data

The data on the population size of Siberian cranes in Poyang Lake span the years 2011 to 2019. Hydrological variables encompass the mean inundated area of Poyang Lake in June, along with the maximum, mean and minimum inundated areas, as well as the rate of dry season recession. The data passed tests for normality and homogeneity of variance, evaluated through the Shapiro-Wilk and Levene tests, respectively. Acknowledging potential multicollinearity among variables, we standardized the data and constructed a correlation matrix. The *p*-value of Bartlett’s sphericity test was 0, indicating significant correlation among variables and suggesting suitability for principal component analysis (PCA). PCA is a multivariate statistical analysis method designed to reduce data dimensionality and reveal patterns within the data [40]. The fundamental idea is to linearly transform the original variables into a set of mutually uncorrelated principal components, where each component is a linear combination of the original variables. We ranked the principal components based on the variance they explained and selected those with higher explained variance. Subsequently, we conducted further regression analysis by choosing the variable with the maximum loading in each selected principal component. The data analysis was performed using the statistical software IBM SPSS Statistics Version 26.0, and the corresponding graphs were plotted using Origin 2021.

### 2.4. Factor Selection and Evaluation Criteria

To analyze the landscape characteristics of Poyang Lake with varying water levels, we established a water level interval of approximately 1 m (Table 1). A total of 10 water levels (Yellow Sea elevation) ranging from 5.3 m to 14.2 m were selected from Xingzi Hydrographic Station. The remote sensing images are derived from Landsat TM/ETM+ data available on the U.S. Geological Survey website (http://glovis.usgs.gov/, accessed on 10 October 2023). Following radiometric calibration and atmospheric correction using ENVI 5.6, the images were clipped based on the vector boundary of Poyang Lake. Utilizing the 4:3:2 false color bands combination, the maximum likelihood method was employed for image interpretation. Based on the habitat requirements of the Siberian crane and the natural landscape types of Poyang Lake, the images were categorized into five groups: deep water, shallow water, grassland, sand, and mudflat. The classification accuracy was assessed through a confusion matrix.

Based on our team’s observations and the previous literature on the wintering habitat selection for the Siberian crane [8,9,17,41], we categorized the environmental factors influencing the habitat suitability of Siberian cranes into three groups: habitat, food and human disturbance. The habitat type data involved the classification mapping of remote sensing imagery, with shallow water classes extracted for Euclidean distance analysis to obtain the distance to shallow lakes (DW). Elevation data (ELE) were extracted from the digital elevation model (DEM) available on the Geospatial Data Cloud Platform (http://www.gscloud.cn/, accessed on 8 October 2023), while NDVI data were sourced from the 250 m normalized vegetation index (NDVI) dataset covering the China region, obtained from the National Tibetan Plateau Science Data Center (https://data.tpdc.ac.cn/home, accessed on 8 October 2023) for the years 2000–2022 [42]. Human footprint index data were acquired from the 2020 dataset on the “figshare” platform (https://figshare.com/, accessed on 8 October 2023), providing information on the impact of human activities on habitats and biodiversity [43]. Each of these variables was standardized in terms of their boundary, coordinate system (WGS_1984_UTM_Zone_50N) and raster resolution (30 m). Following standardization, each evaluation factor was reclassified into four categories, with assigned values of 0–3 representing an unsuitable, poor, fair or good habitat (Figure 2, Table 2). This categorization allowed for a comprehensive assessment of the habitat suitability of Siberian cranes based on these key factors.

#### 2.4.1. Criteria for Evaluating Factor Classification

(a)Habitat factors: Siberian cranes predominantly favor shallow water as their preferred habitat, with mudflats and grasslands also serving as habitats for foraging [8,9,44]. The sand is relatively barren with sparse vegetation, and the deep water makes it difficult for Siberian cranes to reach [45]. The active zone of the crane primarily extends within 500 m of a shallow lake’s edge [9], defining the area within 500 m of shallow water as highly suitable, with suitability decreasing as this distance increases.(b)Food factors: The primary food source for the Siberian crane is the rhizome of wetland plants [12]. Due to the lake’s historical hydrological regime, the vegetation is distributed in a gradient along the lakeside. The submerged vegetation zone preferred by Siberian cranes extends from 10 to 12 m above sea level, while hydrophytes dominated by species such as *Carex* spp. and *Polygonum criopolitanum* are found between 12 and 16 m. Elevations exceeding 16 m are primarily characterized by mesophytic grasslands [46]. Below 10 m, the water level is usually deep and cannot be reached by the Siberian crane. NDVI is often used to assess vegetation conditions [47]. Siberian cranes prefer to inhabit areas with sparse vegetation [44,48].(c)Human disturbance factors: Siberian cranes tend to avoid areas with significant human disturbance [8,48]. The classification of the human footprint index is based on the relevant literature [43] and the actual human disturbance conditions of Poyang Lake.

#### 2.4.2. MaxEnt-HIS Coupling Model

The operation of the MaxEnt model necessitates the input of environmental data and species distribution sites under corresponding environmental conditions. Distribution data of the Siberian crane from December to January 2022 were selected, aligning with an average monthly water level of approximately 8 m. Five environmental variables corresponding to the 7.9 m water level and the active sites of the Siberian crane were included in the model. For modeling, 75% of sites were randomly selected, and the remainder was reserved for model verification. Logistically formatted results were generated after ten repetitions. The accuracy was assessed using the AUC value, ranging from 0 to 1. A value closer to 1 indicates a more favorable outcome, generally considered good when exceeding 0.8 [16,20].

The weights, derived from the contribution rates of different environmental factors modeled using the MaxEnt model, were applied to the reclassified and assigned factors through ArcGIS’s “Weighted Sum” tool for overlay analysis. The results were subsequently normalized to generate the *HSI* layer. Considering that the deep water zones of the main river channel (Figure 1) are unsuitable for submerged vegetation and most hydrophytes’ growth, even in shallow areas during low water levels, this region was deemed unsuitable in the analysis. The boundary was determined by the vegetation distribution map obtained from the comprehensive survey of Poyang Lake, extracted after geographic registration in ArcGIS [45]. The results were then normalized to derive the *HSI* layer, with the calculation formula for the *HIS* being the following:(10)$$HSI=\sum_{i=1}^{n}w_{i}f_{i}$$
where *f_i_* is the *i*-th evaluation factor, *w_i_* is the weight of the *i*-th factor and *n* is the number of evaluation factors.

According to the distribution sites of the Siberian crane, the classification criteria were adjusted, and the habitats were divided into good (0.8 ≤ *HSI* < 1), fair (0.6 ≤ *HSI* < 0.8), poor (0.4 ≤ *HSI* < 0.6) and unsuitable (*HSI* < 0.4) classes [9,17]. Finally, the Siberian crane distribution points obtained in recent years were used to evaluate the prediction accuracy of the MaxEnt-HSI model.

## 3. Results

### 3.1. Annual Variation in the Inundated Area and Receding Rate

The median inundated area of Poyang Lake in the dry season over the last 20 years has exhibited irregular fluctuations, with most displaying negative skewness. The maximum value occurred in 2000 at 2622.70 km^2^, while the minimum was recorded in 2010 at 552.04 km^2^ (Figure 3a), representing a variation of 2070.66 km^2^. In most years, the inundated area remained small (<1500 km^2^), except for instances of larger inundated areas observed in 2002 and 2015. Outliers were notable in 2007, 2009 and 2013. The inundated area demonstrated a fluctuating trend in June (Figure 3b), with the maximum value in 2003 (2829.31 km^2^), the minimum in 2011 (1002.80 km^2^) and the average in 2012 (2729.51 km^2^). In 2000, 2002, 2010, 2012, 2015 and 2016, the inundated area was substantial, exhibiting a predominantly positive skewness.

The interannual variation trends of the maximum, minimum and mean inundated areas were consistent, all exhibiting a downward trend, with changes of 34.53, 14.79 and 8.49 km^2^/a, respectively (Figure 4a, Table 3). Only the maximum inundated area decreased significantly (|*Z*| = 2.26 > 1.96). The rate of water receding did not slow down significantly. It can be seen from the *UF* curve that the maximum, minimum and mean inundated area change trends are similar, mainly showing a downward trend (Figure 4b–d). The maximum area has been decreasing since 2000, while the mean area fluctuated in 2000–2003 and has continued to decline after 2003. The minimum area increased in 2000–2003, continued to decline after 2003 and declined significantly from 2008 to 2015. The *UF* and *UB* curves of the three images have multiple intersections, reflecting the variability in the inundated area, and considering that the curves after the intersection point in 2003 showed a steady downward trend, 2003 was determined to be the mutation point.

### 3.2. Correlation between Siberian Crane Population Size and Hydrological Factors

The correlation heatmap indicates a high correlation among the five hydrological variables (Figure 5a), especially for the four hydrological variables during the dry season. The Bartlett’s sphericity test results for the correlation matrix (*p* = 0) also confirmed that the original dataset was suitable for principal component analysis. The scree plot, depicted in Figure 5b, visually demonstrates the variations in slope for the eigenvalues of the five principal components. Notably, the slope gradually leveled off after the second principal component, as the eigenvalues of the first two principal components were greater than unity and explaining 90.82% of the information contained in the original dataset. Therefore, retaining two principal components was suitable for analysis. The absolute values of loadings reflect the contributions of each variable to the PCs (Figure 5c). In PC 1 and PC 2, the variables with the highest loadings were the mean inundated area during the dry season (B, 0.54) and the mean inundated area in June (A, 0.81), respectively. Therefore, these two variables were selected as dominant factors. To explore the impact of these two variables on the population of cranes, we conducted a multiple stepwise regression analysis. The results showed that in the optimal model, only the mean inundated area in June was retained as a significant factor. It exhibited a significant negative correlation with the crane population (*r* = −0.75, *p* = 0.02, Figure 5d), while the explanatory power of the mean inundated area during the dry season was not significant (*r* = −0.27, *p* = 0.55, Figure 5e). Considering the significant impact of summer flooding on the Siberian crane population, we excluded data from three typical years of summer floods (2012, 2015 and 2016) to focus on the correlation between the crane population and the inundated area in the dry season under normal conditions. We fitted various curve types (linear, quadratic, cubic, etc.) to the remaining dataset, subsequently assessing the quality of the fits. The results indicated that there was a well-fitted quadratic regression equation between the mean inundated area in the dry season and the crane population (*R^2^* = 0.92, *p* = 0.02, Figure 5f).

### 3.3. Suitable Habitat Simulation of Siberian Cranes at Different Water Levels

#### 3.3.1. Habitat Assessment Factor Weights

Based on the confusion matrix method, Poyang Lake landscape classification mapping under different water levels was greater than 90%, and the kappa coefficient was also greater than 0.9. The mean value of the AUC for 10 runs of the MaxEnt model was 0.835 ± 0.065 (mean ± SD), which was a good result. The weights were determined based on the contributions of each evaluation factor: habitat type (0.45), distance from shallow water (0.25), human footprint index (0.15), elevation (0.1) and NDVI (0.05).

#### 3.3.2. Landscape Changes and Suitable Habitat of Siberian Cranes at Different Water Levels

When the water level was higher, deep water dominated (Figure 6), with the maximum area of deep water recorded at 14.2 m (2597.78 km^2^, 73%). The highly suitable habitat for Siberian cranes was the minimum (158.36 km^2^, 4.75%), primarily concentrated in Poyang Lake National Nature Reserve, located in the upper left corner during this period (Figure 7).

With a decrease in water level, the deep-water area gradually diminished, exposing shallow water, grassland, mudflats and sandy beaches. Unsuitable habitats underwent a transformation into fair and poor habitats, whereas good habitats expanded toward areas such as Nanji Wetland National Nature Reserve, Kangshan Migratory Bird Nature Reserve and East Poyang Lake National Park, located in the lower right. Water levels exceeding 12 m resulted in a noticeable decline in suitable habitats (including good and fair suitability). With a decrease in water level, the area of suitable habitats tended to increase, reaching its peak at 5.3 m (1734 km^2^, 52%). The largest area of shallow habitats was reached at 8.8 m (614.37 km^2^, 17%), and simultaneously, good habitats also attained their maximum extent (442.52 km^2^, 13%). Verification with recent Siberian crane distribution data showed that most points were in good and fair habitats, affirming the accuracy of the evaluation results.

## 4. Discussion

### 4.1. Long-Term Hydrological Characteristics of Poyang Lake and Response Mechanism of Siberian Cranes

The water level and inundated area of lakes serve as indicators of the impacts of climate and human activities. Long-term hydrological dynamic studies furnish essential insights that are useful in managing lakes and evaluating Siberian crane habitats [49,50]. Advances in remote sensing technology facilitate the exploration of hydrological dynamics characterized by high variability. This study utilized a spatiotemporal parameter set for surface water based on MODIS NDVI, providing superior temporal accuracy when contrasted with the traditional approach of delineating water bodies using a restricted set of Landsat remote sensing images. This methodology is more adept at monitoring enduring alterations in water bodies and discerning transient variations [28,50].

#### 4.1.1. Hydrological Characteristics of the Dry Season and Its Impact on Siberian Cranes

The interannual variation in the inundated area in Poyang Lake during the dry season illustrates a declining trend in the maximum, minimum and mean values. This suggests a steadfast diminishing trend in the dry season’s inundated area of Poyang Lake over the past two decades, consistent with findings in the pertinent literature [14,51]. The maximum inundated area commonly transpires at the beginning of the dry season, signaling the initiation of the receding period. A notable reduction in the inundated area during this period implies that the early overwintering phase exposes a more extensive portion of the floodland. Over the past two decades, the recession rate of Poyang Lake has displayed a subtle deceleration trend, conceivably associated with the decrease in the maximum inundated area. Poyang Lake encompasses several dish-shaped lakes with a low topography, which, as the water level decreases, gradually detach from the main lake, with some of these sub-lakes being encircled by dykes, thereby artificially impeding their connection with the main lake [52]. The premature exposure of these sub-lakes might contribute to a deceleration in the rate of area change during the water receding process. Concerning the Siberian crane, only a segment of the population migrates to Poyang Lake at the beginning of the wintering period, and the expansion of the exposed area might result in the premature death of submerged plants, thereby diminishing the crane’s food resources in the mid- to late wintering periods [53].

Subsequent analysis revealed multiple abrupt changes in the maximum, minimum and mean inundated area, signifying alterations at various temporal nodes. Taking the mean inundated area as an example, mutation points around 2002 and 2015 might have been associated with abnormal hydrological events in those years, reflecting short-term variations with no enduring impact on the overall trend. However, following the mutation point in 2003, the maximum, minimum and mean inundated areas manifested a protracted downward trajectory, signifying that the 2003 mutation initiated a lasting decline. As a result, the reduction in the inundated area during Poyang Lake’s dry season signified a shift in the hydrological regime, rather than a persistent long-term trend.

The recent drought occurrences in Poyang Lake have attracted considerable attention [14,15]. Changes in the hydrological regime of Poyang Lake are linked to the operations of the Three Gorges Dam (TGD), completed in 2003 and situated in the upper reaches of the Yangtze River [15,54,55]. The construction of the TGD on the upper reaches of the Yangtze River, intended to harness hydropower and mitigate flood and drought disasters in the middle and lower reaches of the basin, first impounded water in 2003. The reservoir’s impoundment resulted in a decline in the water level of the Yangtze River and an increase in the displacement of Poyang Lake. This disrupted the water balance of Poyang Lake and exacerbated the severity of droughts during the dry season [15]. Exacerbated drought is adversely affecting crucial food resources for the cranes, altering the distribution pattern of hydrophytes, where water assumes a vital role. Short-term drought may affect hygrophyte growth, while persistent long-term drought can force vegetation succession [46]. The prolonged drought at Poyang Lake since 2003 has raised the upper limit and reduced the lower limit of distribution thresholds for hygrophytes like *Carex* spp. and *Phragmites australis*. Consequently, this has prompted the contraction of submerged plant distribution, including *Vallisneria*, toward the center of the lake, leading to a substantial reduction in distribution [45,56]. In conclusion, the escalated dry season drought contributes to a scarcity of food resources for Siberian cranes and diminishes the suitability of their natural habitats.

#### 4.1.2. Effects of Extreme Hydrological Events on Siberian Cranes

Over the last two decades, the median fluctuation range of the inundated area during the dry season has been consistently small, primarily displaying negative values, suggesting a relatively stable and concentrated inundated area at lower values. Hydrological dynamics during the dry season are primarily influenced by precipitation and other climatic conditions, with the relatively stable climate making interannual changes not readily apparent [15]. In contrast to the dry season, the inundated area in June manifests pronounced fluctuations, reflecting the climatic instability characteristic of the flood season. Moreover, there are notable extreme values, such as the mean inundated area during the dry season for the years 2000, 2002, 2012 and 2015, all of which markedly exceeded the multi-year average. Similarly, the inundated area in June for 2010, 2012, 2015 and 2016 markedly surpassed the multi-year average, aligning with flood events in the dry and wet seasons, respectively. Significant flood events in Poyang Lake were often linked to the onset of El Niño events, underscoring that climate change, particularly precipitation, played a pivotal role in causing flood disasters [54].

The summer flood in Poyang Lake, especially in June, significantly hampered the growth of *Vallisneria*, frequently resulting in the population’s collapse [13,56]. Our results revealed that the mean inundated area in June for the years 2010, 2012, 2015 and 2016 was the largest observed in nearly two decades. The elevated water level during *Vallisneria*’s growing season diminished submerged light intensity, hindering seed germination, resulting in a decrease in both tuber density and biomass [56]. Monitoring data on *Vallisneria* tubers from the administration of Poyang Lake National Nature Reserve at Dahuchi Lake, Meixi Lake, Shahu Lake and Changhuchi Lake indicated a significant decrease in the density and biomass of *Vallisneria* tubers during these years compared to the multi-year average. Notably, in 2012 (Meixi Lake) and 2016 (Shahu Lake), both the density and biomass of *Vallisneria* tubers reached zero [30,33,34]. The pronounced decline in tubers at Poyang Lake aligned with a noticeable decrease in the Siberian crane population in their natural habitats [13,56].

In 2010, a substantial number of Siberian cranes were initially observed utilizing grassland habitats. Subsequently, the monitored population of Siberian cranes in their natural habitats declined by 2012. From 2015 to 2016, a noticeable shift occurred, with substantial populations of Siberian cranes increasingly observed in artificial habitats [12,13]. Notably, the mean inundated area in the dry season of 2015 markedly surpassed the multi-year average, ranking as the second largest in the past two decades. This might have resulted in the substantial inundation of even grassland habitats, and the marked loss of natural habitats prompted Siberian cranes to transition toward artificial habitats. In 2012, the number of Siberian cranes in their natural habitat resembled that of 2015 (around 2000, below the multi-year average of 2722) [30,33], and the extent of inundation during the dry season was also substantial. During this period, certain foraging habitats of Siberian cranes might have experienced a shift. Despite the dry season’s inundated area in 2016 returning to the annual average, Siberian cranes did not choose to forage in grasslands, as observed in 2010. Instead, they extensively utilized artificial habitats such as paddy fields and lotus ponds [13].

The lotus pond in the Wuxing reclamation farm serves as a significant artificial habitat for Siberian cranes. Over the recent years, the recorded number of Siberian cranes has steadily increased, reaching 2900 by 2020 [57]. This upward trend suggests that artificial habitats offer certain advantages as a supplementary food source, providing relatively rich food resources and posing fewer feeding challenges [58]. Despite the rebound in *Vallisneria* abundance within natural habitats, the presence of these advantages consistently prompts certain Siberian cranes to choose artificial habitats. However, the persistent high level of disturbance remains a critical factor limiting the majority from utilizing artificial habitats [13,25,57]. Since Siberian cranes began utilizing artificial habitats due to summer flooding, these habitats have evolved into a vital type of habitat for them. Depending on water levels, Siberian cranes can adaptively adjust their proportion of use between natural and artificial habitats to enhance their overall suitability [59]. In summary, the dominant factors influencing foraging habitat selection by Siberian cranes were the abundance of food and the availability of foraging habitat [58].

#### 4.1.3. Correlation between Siberian Crane and Hydrological Characteristics in Poyang Lake

The results of the principal component analysis indicated that the cumulative variance explained by the first two principal components was 90.82%, emphasizing their efficacy in capturing the variability in the original data. Among these components, the highest loading values corresponded to the inundated area in June and the dry season, identified as dominant factors. Regression analysis revealed a relatively weak correlation between the population size of Siberian cranes in the natural habitat of Poyang Lake and the mean inundated area during the dry season. Nevertheless, a noteworthy negative correlation with the mean inundated area in June implies that a larger inundated area significantly contributed to the decline of Siberian cranes in natural habitats. The summer floods in 2012, 2015 and 2016 resulted in a reduction of about 1000 cranes in natural habitats compared to normal years. Elevated water levels during summer caused a substantial reduction in *Vallisneria* density, significantly diminishing natural habitat suitability and reducing the carrying capacity for Siberian cranes [13].

After excluding data from years with summer floods, we observed a well-fitted quadratic curve between the mean dry season’s inundated area and the population size of the cranes. When the inundated area was either large or small, the number of Siberian cranes decreased, whereas moderately sized areas could sustain a larger population of Siberian cranes. Specifically, the mean inundated area in the dry season was smaller in 2011 and 2013, and larger in 2017 and 2018. In both cases, there were fewer Siberian cranes compared to 2014, which had an area of moderate size. This means that in years with normal summer water levels or, in other words, with abundant *Vallisneria* tubers, the impact of the dry season’s inundation on the number of cranes became apparent, acting through its influence on the availability of food for the cranes. An excessively small inundated area may lead to a significant death of submerged vegetation, while an excessively large inundated area implies a higher proportion of deep-water areas inaccessible to cranes. However, the reduction in Siberian crane numbers was not as significant as in the years with summer floods. Research suggests that Siberian cranes can adapt to varying hydrological conditions by adjusting the area of their activity range [59]. In years with normal summer water levels, Siberian cranes find ample food in their natural habitats, satisfying most of their energy needs. Consequently, they do not need to spread out in search of sustenance, resulting in stable population numbers [59]. In contrast, during years of elevated summer water levels, Siberian cranes need to extend their activity area to find appropriate foraging grounds, despite the dry season’s inundated area being suitable. The scarcity of food leads to a reduction in the Siberian crane population within their natural habitat [13,59].

### 4.2. Impact and Importance Analysis of Environmental Factors on Habitat Selection by Siberian Cranes

To comprehensively assess the impact of the dry season’s hydrological conditions on the Siberian crane’s habitat quality, this study quantified habitat suitability under various water level scenarios using the MaxEnt-HSI model. The selection and weighting of environmental factors are pivotal steps in HSI modeling [17]. Siberian cranes are highly dependent on shallow water wetlands, with both their wintering and migratory stopover sites located in shallow areas rich in hygrophytes [60,61]. Their preference for shallow water habitats aligns with their morphological characteristics, which include a longer bill and tarsus, suitable for wading and digging for food [8]. The presence of water in the lake aids in cleaning the sediment carried by plant roots and stems, and the softer substrate under the water enhances the foraging efficiency of Siberian cranes [44]. For instance, at the Jilin Momoge stopover site, Siberian cranes predominantly feed on the roots of *Typha orientalis* and *Cirpus planiculmis*, both of which grow in shallow water and mudflats. However, the cranes rarely feed on mudflats, underscoring the importance of the water environment for Siberian cranes [62]. Therefore, in this study, we selected the habitat type and distance to shallow lakes as the two habitat factors representing the preference of Siberian cranes for a water environment. These factors substantially influence the survival and distribution of Siberian cranes, and the MaxEnt model confirmed that they bear the greatest weight.

Human disturbance plays a crucial role in evaluating the habitat suitability of species and frequently has a detrimental impact on wildlife [63]. Siberian cranes exhibit a high sensitivity to human disturbance, typically taking flight when humans approach within 200 m of them [44]. Previous studies evaluating habitat suitability have often employed distance to disturbance sources, such as roads and villages, as a factor for assessing human disturbance. This paper employed the human footprint index (HFI) dataset, considering eight variables closely related to human activities, including population density, built environment, roads and navigable waterways, to enable a more comprehensive characterization of the intensity of human disturbance [43]. Siberian cranes exhibit a preference for shallow water habitats and their surrounding areas, usually located away from human activity, showcasing a high sensitivity to human interference. Consequently, the weight assigned to human disturbance factors was relatively high.

Food is a vital consideration in the choice of habitats for waterbirds, especially during the overwintering period. The abundance and quality of food directly impact energy storage, which in turn affects migration and breeding processes [8]. The Siberian crane exhibits a strong specialization in food selection, primarily favoring the tubers of *Vallisneria*, a dominant submerged plant in shallow lakes, being its preferred food source [12]. In the absence of *Vallisneria* at high-water levels or when their biomass and density are low due to extreme weather, the crane also feeds on hygrophytes such as *Potentilla limprichtii* and *Polygonum criopolitanum* [12,64]. Since hygrophytes are typically mixed in a community, and submerged plants are challenging to identify in remotely sensed images, interpreting vegetation type as a food factor becomes difficult [9]. Considering that the wetland vegetation in Poyang Lake is influenced by factors like water level fluctuations and soil moisture forming different community types along elevations [65], this study employed elevation as an indicator for the distribution of vegetation types. Figure 2a illustrates that the area corresponding to an elevation range of 10–12 m predominantly represents shallow water, aligning with the actual distribution of *Vallisneria*, indicating that the elevation gradient can, to some extent, characterize the distribution of vegetation. Additionally, this study utilized NDVI data to represent vegetation cover, since Siberian cranes exhibit a preference for sparsely vegetated habitats, with higher vegetation density tending to impede their foraging [44,60]. Owing to the relatively small geographical scale of the Poyang Lake area, which results in minimal elevation variation across the region, the influence of the elevation factor may be comparatively limited. Although vegetation coverage does affect crane habitat selection, its impact is relatively weaker compared to other dominant factors, such as habitat type, resulting in the smallest weight assigned to it.

In summary, the five evaluation factors selected in this study encompassed the three levels of habitat, food and human disturbance, effectively representing the ecological needs of Siberian cranes.

### 4.3. Suitable Habitat Distribution Pattern of Siberian Crane at Various Water Levels

The HSI model indicated that good habitats for Siberian cranes are primarily situated in dish-shaped lakes and their surrounding areas, characterized by rich submerged vegetation, with less anthropogenic interference and relatively wet substrate conducive for the crane’s foraging. This preference enables the crane to efficiently access a substantial amount of food resources, aligning with the optimal foraging theory [12]. The fair habitats include mudflats and grasslands near a dish-shaped lake, remaining relatively wet during the short period following water recession. Grasslands may also begin budding, providing some food resources for Siberian cranes. The poor habitats include mudflats, grasslands and sections of sandy beach distant from shallow lakes, which remain dry for an extended period. In contrast, the unsuitable zone comprises sandy beaches and deep-water lakes. Sandy beaches primarily located in the Songmen Mountain and Shashan areas of Dobao [41], gradually emerge as the water level drops. Over time, some of these sandy beaches transform into grasslands, exhibiting a trend of increasing and then fluctuating changes. The sandy beach substrate is unsuitable for the growth of submerged vegetation and most hydrophytes, lacking the necessary food resources for cranes; consequently, it is considered an unsuitable habitat for crane habitation.

When the water level reached 14.2 m, deep water became the predominant landscape type, and only a few dish-shaped lakes with higher topography in the northwest of Poyang Lake National Nature Reserve were exposed. Consequently, there were fewer good habitats, and Siberian cranes were primarily distributed around the dish-shaped lake to feed on the buds or rhizomes of hygrophytes [44]. As the water level declined, mudflats were extensively exposed, and Siberian cranes predominantly foraged in the zone of submerged and mixed plants during this period [44,53], leading to an enlargement of the area with good habitats. With further water level reduction, the dish-shaped lake was fully exposed, reaching a water depth suitable for Siberian cranes to feed on. At this point, the cranes mainly fed on the rhizomes of submerged plants such as *Vallisneria*, *Potamogeton malaianus*, etc. [53]. In this period, Siberian cranes can make the most use of their habitat, with the area of good habitat reaching its maximum, corresponding to a water level of 8.8 m. Subsequently, as the water level continued to drop, the dish-shaped lake contracted toward its center, diminishing in size and resulting in a decrease in good habitats for the cranes. Prolonged digging and feeding in the early stages also led to a decline in food resources in shallow water habitats, as increased exposure time hardened the substrate in mudflats, and severe plant fibrosis in grasslands reduced palatability [66], making it less conducive to Siberian crane feeding. Hence, the habitat at this stage was mainly categorized as fairly and poorly suitable.

It is noteworthy that the area of unsuitable habitats tended to decrease significantly with the decline in water level, reaching the smallest area at a 5.3 m water level. Conversely, good habitats initially increased and then decreased with the decreasing water level. The suitability was higher in the 8–10 m water level range, where medium water levels provided more good habitats for Siberian cranes, whereas low and high water levels were relatively unsuitable for their survival, further confirming the correlation between the dry season’s inundated area and the crane’s population size. Previous research based on population growth models has also indicated that maintaining a moderate water level within the range of 8.7–10.2 m during the dry season is beneficial for the growth of the Siberian crane population [67]. However, despite maintaining the water level within an optimal range for the ecosystem, which provides a larger and highly suitable habitat, extensive foraging by the Siberian crane and other geese species, such as *Cygnus columbianus*, which feed on *Vallisneria*, gradually reduces the environmental carrying capacity [59]. This leads to a shortage of food supply in the later stages of wintering. Therefore, in the actual process of water level adjustment, it may not be sufficient to only consider the size of the suitable area; it is also necessary to monitor the real-time dynamics of the crane population and food resources.

### 4.4. Protection and Management

The alteration in hydrological characteristics in Poyang Lake underscores its ecological vulnerability, contributing to a decline in habitat quality for Siberian cranes. Therefore, meticulous habitat management aligned with the ecological requirements of Siberian cranes is imperative. Based on this study, the following recommendations are proposed. (a) Adjustment of water level: Modify water level regulations by considering the actual hydrological conditions of downstream rivers and lakes during water storage and drainage, with the aim of alleviating summer floods and winter droughts. To optimize food supply for Siberian cranes, gradually lower the water level before the middle of the wintering period, maintaining an optimal ecological depth of approximately 8–10 m when crane populations peak. Subsequently, continue decreasing water levels during the late wintering phase to expose submerged vegetation in deeply inundated areas at the center of dish-shaped lakes. (b) Restore wetland ecology and accelerate vegetation restoration: Enhance the ecological environment of Poyang Lake and revive submerged vegetation. This can be achieved by improving water quality, promptly adjusting water levels and implementing artificial planting. (c) Strengthen monitoring: Conduct long-term monitoring of Poyang Lake wetland vegetation and Siberian crane populations to comprehend their fluctuations. Perform countermeasure analysis and early warning systems based on observed changes. (d) Enhance artificial habitat management: Artificial habitats can offer relatively stable foraging grounds for the cranes, especially when natural habitats lack sufficient food resources.

## 5. Conclusions

In order to assess the changes in the habitat quality of the Siberian crane at Poyang Lake over the last 20 years, we conducted a comprehensive analysis of the hydrological characteristics of Poyang Lake. This analysis involved the application of the Mann-Kendall and Sen slope trend tests to investigate the response mechanism of Siberian cranes to the changes in the long-term hydrological trend during the overwintering period in Poyang Lake. Subsequently, employing the coupled MaxEnt-HSI model, we simulated the distribution of suitable habitats for Siberian cranes under different water level scenarios. The following results were obtained: (a) Over the past 20 years, the inundated area of Poyang Lake exhibited a general decreasing trend during the dry season, with the maximum inundated area significantly reduced, leading to a slower rate of water recession. (b) The change in the inundated area of Poyang Lake during the dry season signifies a shift in the hydrological regime, with a noticeable change point in 2003. (c) The summer floods resulted in a decrease in the abundance of *Vallisneria*, driving a shift in foraging strategies and habitat changes for Siberian cranes. The prolonged trend of drought exacerbated the shortage of food for Siberian cranes. (d) The mean inundated area of Poyang Lake in June primarily caused fluctuations in the population of Siberian cranes by influencing the amount of food resources in their natural habitat. Additionally, the inundated area during the dry season predominantly affects the food availability for Siberian cranes, consequently influencing their population size and distribution pattern. (e) The area of good habitat for Siberian crane increased and then decreased with the decrease in water level, the suitability was higher in the 8–10 m water level range, and the negative effects of a high water level were higher than those of a low water level. Overall, this study concludes that the hydrological characteristics of Poyang Lake in recent years will further reduce the suitability of Siberian crane habitats, and food shortage may become the new normal. The degradation of natural habitats may lead to an increased reliance on artificial habitats by the Siberian crane. Therefore, striking a balance between the restoration of natural habitats and the management of artificial habitats is crucial for the conservation of Siberian cranes.

Although our study has made progress in assessing the alterations in habitat quality for Siberian cranes and their response mechanisms to changes in the hydrological regime in Poyang Lake over the preceding two decades, it is imperative to candidly recognize certain limitations in our study. Firstly, our research relied on existing data and available resources, leading to a relatively modest sample size. This constraint may have somewhat limited our comprehensive understanding of the population-level response of Siberian cranes to hydrological changes. Secondly, despite our endeavors to incorporate a spectrum of hydrological characteristics in our analysis, our variable selection might not have encompassed all potentially more representative influencing factors, such as the duration of suitable inundation areas. These limitations emphasize the imperative for future research to contemplate a broader spectrum of factors and, where feasible, augment sample sizes to attain a more profound and comprehensive understanding of Siberian cranes’ response to hydrological changes in Poyang Lake.

## Figures and Tables

**Figure 1 animals-14-00234-f001:**
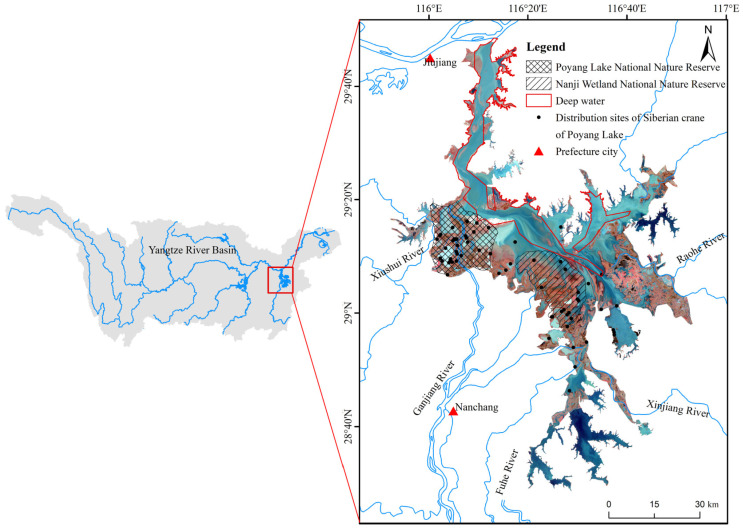
Geographical mapping of the Poyang Lake wetland, national nature reserves and Siberian crane distribution.

**Figure 2 animals-14-00234-f002:**
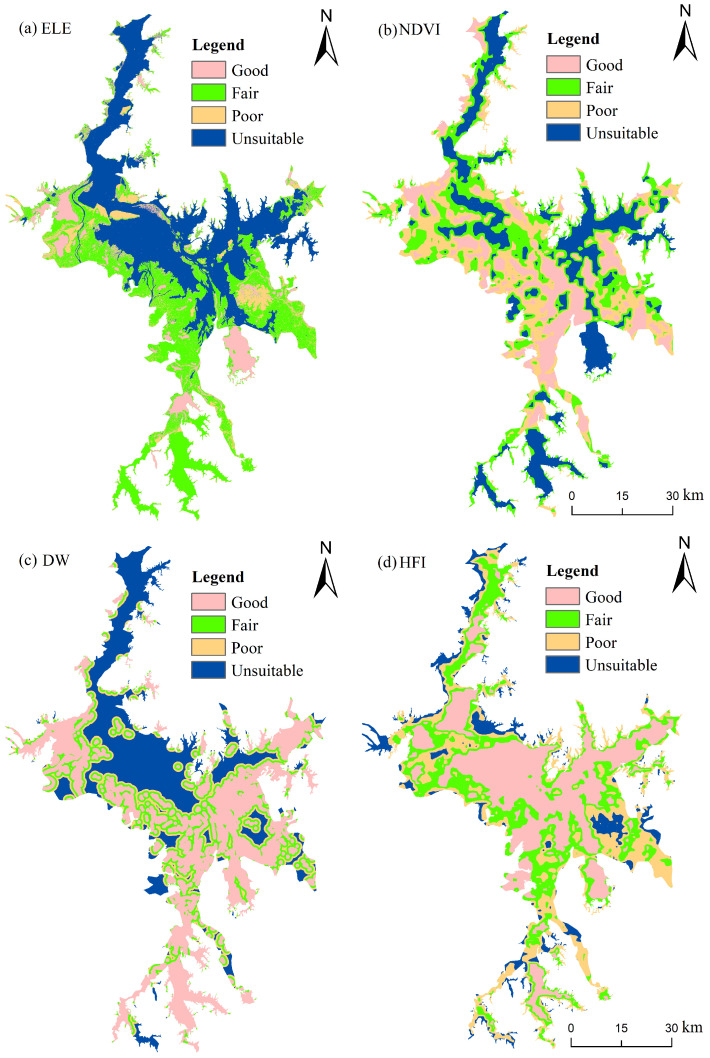
Factors influencing habitat suitability for Siberian cranes: (**a**) elevation (ELE), (**b**) NDVI, (**c**) distance from shallow lakes (DW) and (**d**) human footprint index (HFI). Note: (**b**,**c**) correspond to a water level of 10.1 m.

**Figure 3 animals-14-00234-f003:**
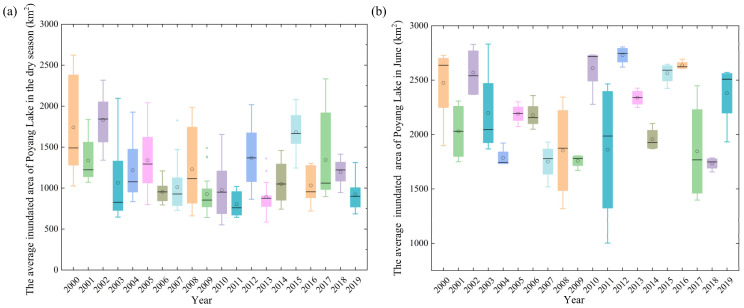
Box plots of the mean inundated areas in (**a**) the dry season and (**b**) June of Poyang Lake. The horizontal lines of the boxes represent the medians and the black circles inside the boxes represent the means. The lengths of the boxes symbolize the interquartile ranges (IQRs), with the box ends corresponding to the upper quartile (Q3) and lower quartile (Q1). The upper and lower whiskers connect to the maximum and minimum values, respectively. Colored dots outside the boxes denote outliers, exceeding Q3 + 1.5 × IQR or falling below Q1–1.5 × IQR.

**Figure 4 animals-14-00234-f004:**
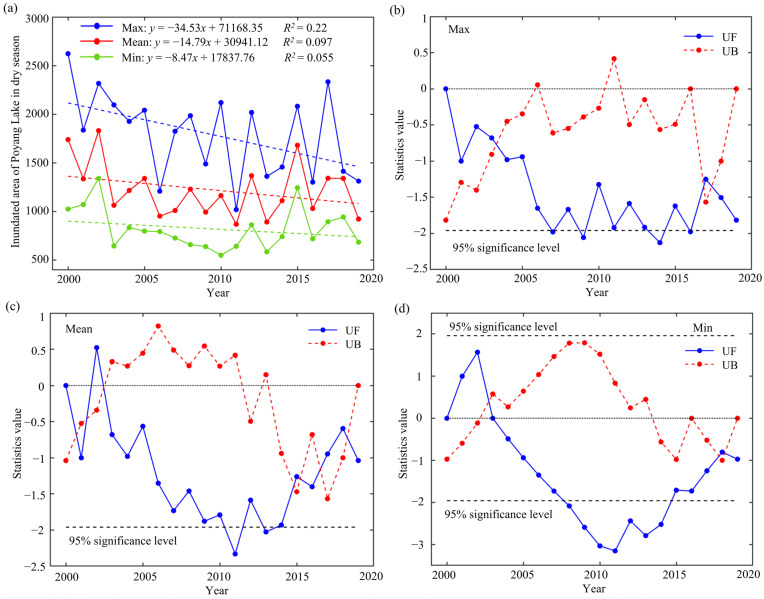
Trends in the temporal variation in Poyang Lake’s inundated area during the dry season and the outcomes of abrupt change point detection. (**a**) Illustrated trends in the maximum, mean and minimum inundated areas during the dry season from 2000 to 2019, the dashed lines of various colors illustrate the trend lines for the corresponding color data. Outcomes of abrupt change point detection for the (**b**) maximum, (**c**) mean and (**d**) minimum inundated areas. *UF* curve values greater than 0 signify an upward trend, while values less than 0 indicate a downward trend, and values exceeding ±1.96 (95% significance level, shown by black dashed lines in **b**–**d**) indicate a significant trend. The point where the two curves intersect within the critical range signifies the abrupt change point.

**Figure 5 animals-14-00234-f005:**
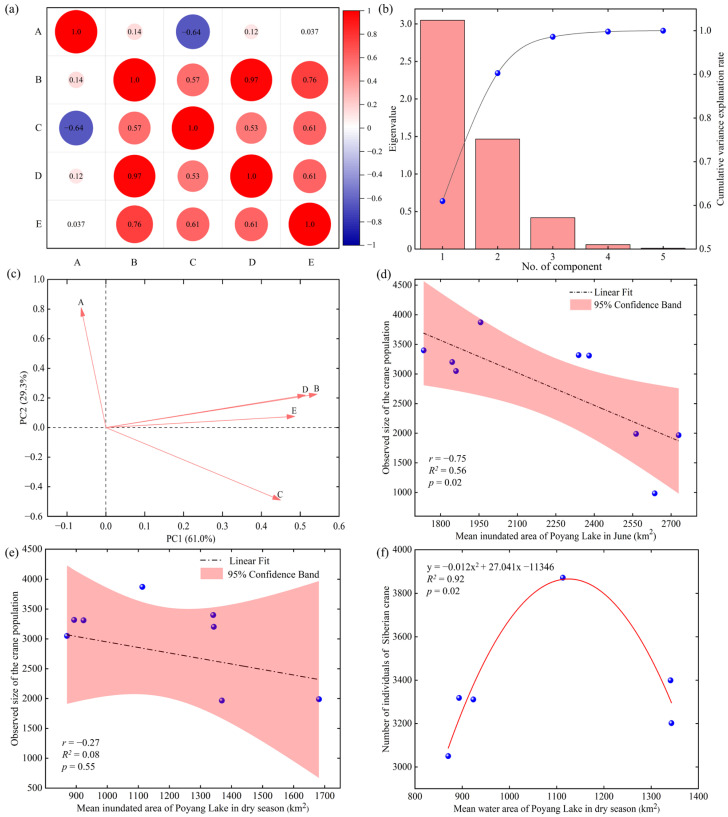
Overview of principal component regression analysis between the five hydrological variables in Poyang Lake, spanning from 2011 to 2019, and the crane population. (**a**) Heatmap of Pearson’s correlation analysis for the five hydrological variables (A, B, C, D and E), representing the mean inundated area in June, as well as the mean, maximum and minimum inundated areas during the dry season and the water recession rate of Poyang Lake. (**b**) Dual-axis graph illustrating PCA eigenvalues and the cumulative variance explained by each component. The bar chart depicts the PCA eigenvalue scree plot, whereas the line graph illustrates the cumulative variance explanation rate of each component. (**c**) Loading plot depicting the five hydrological variables. The projection of each variable onto the coordinate axes represents its loading value in the corresponding principal component. (**d**,**e**) Regression analysis plots between the two variables with the highest loading values in principal components 1 and 2 and the observed size of the crane population. (**f**) Polynomial regression curve between the mean inundated area during the dry season and the population of cranes after excluding years with an abnormally high mean inundated area in June.

**Figure 6 animals-14-00234-f006:**
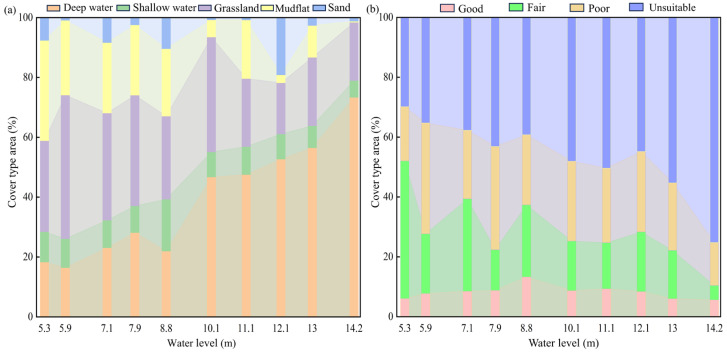
(**a**) Proportions of different landscape types and (**b**) suitable habitat areas for Siberian cranes at various water levels in Poyang Lake.

**Figure 7 animals-14-00234-f007:**
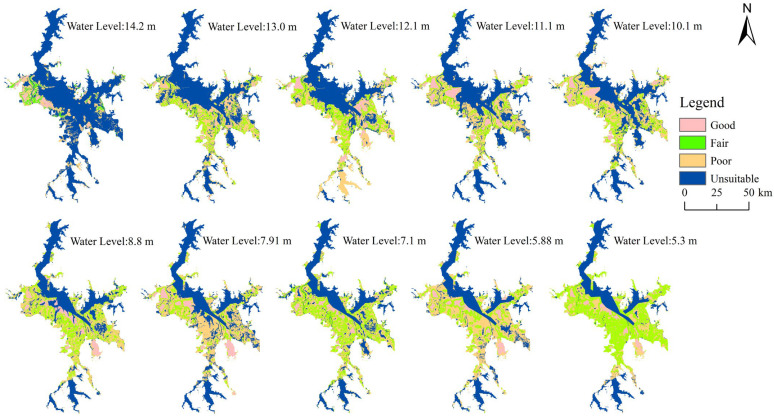
Distribution patterns of suitable habitats for the Siberian crane at various water levels.

**Table 1 animals-14-00234-t001:** Acquisition time of TM/ETM+ images and corresponding water levels at Xingzi Hydrological Station.

Image’s Acquired Date	Sensor	Water Level at Xingzi Station
(DD/MM/YYYY)		(Yellow Sea Elevation)
15 February 2004	TM	5.3
6 January 2007	ETM+	5.88
15 December 2004	TM	7.1
27 January 2000	TM	7.91
10 December 1999	ETM+	8.8
5 March 2005	TM	10.1
16 November 1999	TM	11.1
2 November 1994	TM	12.1
5 October 2007	TM	13.0
9 October 2000	ETM+	14.2

**Table 2 animals-14-00234-t002:** Suitability evaluation criteria for ecological factors.

Factors	Evaluation Factor	Good (3)	Fair (2)	Poor (1)	Unsuitable (0)
Habitat factors	HT	Shallow water	Mudflat	Sand	Deep water
			Grassland		
	DW	<500 m	500–1000 m	1000–1500 m	>1500 m
Food factors	ELE	10–12 m	12–14 m	14–16 m	<10 m and >16 m
	NDVI	0–0.1	0.1–0.3	0.3–0.5	0.5–1
Human disturbance factors	HFI	0–6	6–12	12–20	>20

**Table 3 animals-14-00234-t003:** Interannual variation rate and trend analysis of hydrological variables employing the Mann-Kendall test.

Statistical Value	Maximum Inundated Area	Mean Inundated Area	Minimum Inundated Area	Recession Rate
Rate of change	−34.53	−14.79	−8.47	−2.01
*Z* value	−2.26 *	−1.54	−1.47	−0.75

Note: A positive *Z* value indicates an upward trend, while a negative *Z* value indicates a downward trend. If the absolute value of *Z* (|*Z*|) exceeds 1.96, the trend is considered significant and is marked with an asterisk.

## Data Availability

The data presented in this study are available upon request from the corresponding author. The data are not publicly available due to the constraint in the consent.

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
