# Peer review of "Response of Siberian Cranes (Grus leucogeranus) to Hydrological Changes and the Availability of Foraging Habitat at Various Water Levels in Poyang Lake"

_animals, 2024, doi:10.3390/ani14020234_

Round 1

Reviewer 1 Report

Comments and Suggestions for Authors

Dear authors,

Your study is interesting and important, but the results are difficult to follow. Please consider these comments below as you make revisions.

1.       I recommend replacing “water area” with “inundated area” throughout the manuscript.

2.       It is not clear to me whether the simple summary and the abstract convey the same main results. With respect to the observed shift of crane habitat, is the issue the decrease in the total inundated area during winter (Lines 14-15) or is it the June floods (Line 30)? Unfortunately, this discrepancy sets the tone for the rest of the paper, where it is not always easy to follow you as you present and discuss your findings. 

3.       Lines 62-63. I would indicate that this cycle repeats itself every year.

4.       Lines 69-72 (and elsewhere). You mention 2003 as a "mutation point" in the results. As I understand it, 2003 is when the Three-Gorge Dam on the Yangtze River began to fill. Why not mention that dam as one or the main reason for hydrological changes at Poyang Lake? I urge you to incorporate some of the results from the recently published paper (available online since May 2023), linking a hydrological regime shift in low water level variations with the Three Gorge Dam:

Li, B., Yang, G., Wan, R. Reassessment of the declines in the largest freshwater lake in China (Poyang Lake): uneven trends, risks and underlying causes.  Journal of Environmental Management 2023, 342, 118157.

5.       Lines 158-160, “Water level data of Poyang Lake in 2021-2022 was obtained from Jiangxi Provincial Water Resources Department (http://slt.jiangxi.gov.cn/).” I recommend moving this sentence to the end of the previous paragraph, which discusses water level data instead of crane population data.

6.       Lines 243-246, you might want to indicate this elevational zonation happens to be the result of the lake’s historical hydrological regime (underwater gradients would typically be determined by depth rather than elevation per se).

7.       Figure set 5, Y axis title of first figure is different from that of the other figures. It should probably be, “Observed size of the crane population” (as should be the case for the others as well).

8.       Figures 5b and 5c appear to be duplicates of each other!

9.      Is it possible to say anything about the proposed drought-mitigation sluice wall proposed by the Jiangxi provincial government but apparently opposed by environmental NGOs? Would it help the cranes and their foraging habitat? Are you in favor of the sluice wall when you recommend that (lines 616-25) “[… the water level regulation needs to be adjusted […] to alleviate the flood in summer and drought in autumn and winter. To maximize the food supply for Siberian crane, it is advisable to gradually lower the water level before the middle of the wintering period, aiming to maintain an optimal ecological depth of approximately 8-10 meters when crane populations reach their peak. Following this, a continued decrease in water levels during the late wintering phase to expose the submerged vegetation in the deep-water areas at the center of dish-shaped lakes.”

Good luck with the revisions!

Comments on the Quality of English Language

The manuscript definitely needs to be edited

Author Response

Please see the attachmet.

Reviewer 2 Report

Comments and Suggestions for Authors

The manuscript entitled “Response mechanism of Siberian cranes (Grus leucogeranus) to changes in hydrological characteristics and simulation of suitable habitat at different water levels in Poyang Lake” by Mingqin Shao and colleagues, submitted to the Animals magazine, presents results of a study on hydrological data, and the relationship between water levels in Poyang Lake and the crane populations. The Siberian crane is an endangered species, and wetlands are important habitats for the bird. Thus, this study is important and interesting; however, I think that this version of manuscript could be improved.

General remarks
Authors analysed data on hydrological conditions at the Poyang Lake from 2011 to 2019, and population size of the crane. I believe that the most important aim of the analyses was to try to find which of the analysed hydrological data are corelated with the Siberian crane population size (Authors analysed also some environmental factors affecting the habitat for the crane like: habitat, food and human disturbance, but I believe that the changes in hydrological characteristics are essential for the study).
The existence of a correlation doesn’t prove causation. However, it could be the first step for trying to understand analysed data, analysed processes etc. Thus, I understand why the Authors used such analyses, but I think that there is problem with the analyses.

See for example:
(1)
-- Figure 5 a-e. The relationship between Siberian crane population size and water areas are presented. However, from the five datasets presented on figures 5(a)–5(e), the correlation for dataset presented on the figure 5(a) is significant only,
and
-- lines 326-317: “The population size of Siberian crane in the natural habitat of Poyang Lake was significantly negatively correlated with the mean water area in June (r=-0.75, p<0.05)”. The exact p value should be presented here (“p<0.05” is not enough). Additionally, if several correlation coefficients were calculated, the Bonferroni correction should be used for such analyses, I think. In such situation (i.e., when the Bonferroni correction will be used), the correlations will be not statistically significant, I suppose.

(2)
-- lines 320-321: “The relationship between the population size of Siberian cranes and the mean water area in the dry season showed binomial fitting curve and had a good fitting effect (R2=0.92).”
There is lack of p value for the analysis.
Additionally:
-- lines 482-483: “Our results showed that the relationship between the two was better fitted as a binomial curve.”;
however, the curve is based on just one point for mean water area between the values c.a. 1000 and 1300 km2. Thus, I am not sure, if the relationship is statistically significant.

To sum up: There is problem with the statistical analyses and/or presentation of results of the analyses, and thus with interpretation of the main results of this study.

Some other comments:
-- I feel that some part of manuscript could be prepared in the better way. For example, the sentence (lines 377-380) “The interannual variation of water area in Poyang Lake during the dry season showed a decreasing trend among the maximum, minimum, and mean values, indicating that the dry season water area in Poyang Lake has shown a shrinking trend in the last 20 years, which is in agreement with what has been reported in the relevant literature [14,50].” is quite long, and – for me – difficult to follow.

-- 2.3.1. Detection of Changes in Water Area
It looks like part of textbook to statistics: “Suppose the time series is random, define the statistic “, In reverse order of the time series X(Xn, Xn-1, . . ., X1)repeat the above proces”, “Take the significance level” etc.
I would like to read why this method was chosen, not how to calculate statistics – for this cite any good textbook.  

-- 2.3.2. Correlation between Siberian Crane Population size and Hydrological Data
More information is necessary. Have you used Pearson correlation coefficient? Have you checked assumptions for it? If yes – it should be stated.

-- All used symbols should be explained. For example, what is “±” (line 334: “l0.835±0.065”)? SE? SD? ..?

Figures and tables:
-- Generally, better captions of figures and tables, would be useful for readers. In scientific papers, captions of figures and tables should be ‘self-explaining’, i.e., should provide sufficient information to the readers without looking for information in the text. Thus, for example, the legend “Table 3. Interannual rate of change and z-value.” is not enough, I think [additionally, in the table 3, more data could be presented, I believe; p values? The results are important for the study, as one of the Z values from the table is presented in the Abstract section].
The legend of the Figure 4 could be prepared in a better way, e.g., there are no explanation of abbreviations used on the Figure and the figure legend. Thus, I am not sure, if I properly understand this figure. 

-- Several figures could be prepared in a better way, for example:
Figure 2. Why the same colours are used for different parameters? See for example ‘green’ – on the (a) means ‘fair’, on (b) and (c) ‘poor’; ‘purple’ – on (a) ‘good’, but on (d) ‘poor’. It could be confusing for readers.
Figure 3. Why Y axis for (a) present range ‘0-3000’, but for (b) range ca. ‘500-3000’? Why the average is presented on the (b) only? I have problem to understand it. 

To sum up: more information in captions/legends in some other figures and tables would be useful for readers.

‘Institutional Review Board Statement’ – should be correctly added, or this part removed from the text.
‘Data Availability Statement’– appropriate information should be added.

Author Response

Please see the attachmet.

Round 2

Reviewer 1 Report

Comments and Suggestions for Authors

Dear authors,

The manuscript is much improved. Thank you for addressing my comments. My two comments on the revised manuscript are:

1) The title is now misleading. I suggest instead: "Response of Siberian cranes (Grus leucogeranus) to hydrological changes and the availability of foraging habitat at various water levels in Poyang Lake". Otherwise, your title now suggests that you are looking at the mechanisms through which Siberian cranes are adapting to varying hydrological conditions and water levels.

2) The Y axis title for Figure 6 needs to be corrected from "Area ration (%)" to something like "cover type area (%)" (a ratio is different from a percentage)

And note also a typo in the legend above, where it should be "grassland" instead of "grasslan"

That's it. Well done!

Comments on the Quality of English Language

Some editing will be necessary in English (including of figure axis titles)

Reviewer 2 Report

Comments and Suggestions for Authors

The manuscript has been improved and this version is significantly better than the previous one.

Still, I am not sure if using a binomial curve for data analysis is correct (see e.g. Lines 559-600 “Our results reveal that the relationship between the two variables is best fitted as a binomial curve”).
I understand that the result is statically significant; however, why for data presented on the figures 5d and 5e the linear model was used, but for the data presented on the Figure 5f – binomial curve? (And again: the curve is based on just one point for mean water area between the values c.a. 1000 and 1300 km2.) I think, it should be accurately explained in the manuscript. Thus, more information in the area, in the Material and method section (as well as in the Discussion section), is recommended.

Line 379: “the crane population (R2 = 0.92, r = 0.02, Figure 5f).” ‘r = 0.02’ or ‘P = 0.02’?
